# *SiNF-YC2* Regulates Early Maturity and Salt Tolerance in *Setaria italica*

**DOI:** 10.3390/ijms24087217

**Published:** 2023-04-13

**Authors:** Jiahong Niu, Yanan Guan, Xiao Yu, Runfeng Wang, Ling Qin, Erying Chen, Yanbing Yang, Huawen Zhang, Hailian Wang, Feifei Li

**Affiliations:** 1Featured Crops Engineering Laboratory of Shandong Province, National Engineering Research Center of Wheat and Maize, Crop Research Institute, Shandong Academy of Agricultural Sciences, Jinan 250100, China; 2College of Life Science, Shandong Normal University, Jinan 250358, China

**Keywords:** *Setaria italica*, QTL mapping, *SiNF-YC2*, early maturity, photoperiod sensitivity, abiotic stress

## Abstract

Early maturity is an important agronomic trait in most crops, because it can solve the problem of planting in stubble for multiple cropping as well as make full use of light and temperature resources in alpine regions, thereby avoiding damage from low temperatures in the early growth period and early frost damage in the late growth period to improve crop yield and quality. The expression of genes that determine flowering affects flowering time, which directly affects crop maturity and indirectly affects crop yield and quality. Therefore, it is important to analyze the regulatory network of flowering for the cultivation of early-maturing varieties. Foxtail millet (*Setaria italica*) is a reserve crop for future extreme weather and is also a model crop for functional gene research in C4 crops. However, there are few reports on the molecular mechanism regulating flowering in foxtail millet. A putative candidate gene, *SiNF-YC2*, was isolated based on quantitative trait loci (QTL) mapping analysis. Bioinformatics analysis showed that *SiNF-YC2* has a conserved HAP5 domain, which indicates that it is a member of the NF-YC transcription factor family. The promoter of *SiNF-YC2* contains light-response-, hormone-, and stress-resistance-related elements. The expression of *SiNF-YC2* was sensitive to the photoperiod and was related to the regulation of biological rhythm. Expression also varied in different tissues and in response to drought and salt stress. In a yeast two-hybrid assay, *SiNF-YC2* interacted with *SiCO* in the nucleus. Functional analysis suggested that *SiNF-YC2* promotes flowering and improves resistance to salt stress.

## 1. Introduction

Early maturity is an important agronomic trait in most crops. It can solve the problem of planting in stubble for multiple cropping, improve the multiple cropping index, and increase the annual crop yield [1]. In particular, in high-latitude, high-altitude alpine regions with short frost-free periods, early maturity can make full use of light and temperature resources while also avoiding the damage caused by low temperatures in the early growth period and early frost in the late growth period to improve crop yield and quality [2]. Thus, in alpine regions, early maturity is important for safe crop production. The practices carried out in alpine regions of China are mostly a combination of agriculture and animal husbandry, with seriously deficient accumulated temperature and drought with little rain. The lack of effective crops with drought tolerance that provide both grain and grass is one of the main factors restricting the development of local economies. As a traditional characteristic crop in China, foxtail millet is drought-tolerant and can be used to provide both grain and grass, and therefore, it is suitable for development in arid and water-scarce alpine regions. However, because of the lack of early-maturing varieties that can mature normally in local areas, millet is difficult to plant in the alpine regions of China, restricting millet planting in those areas [3]. The cultivation of early-maturing millet varieties is the most suitable solution to the above problems. Thus, to expand the suitable planting range of millet, it is urgent to improve crop breeding and incorporate genes associated with early maturity. The discovery of early-maturity genes is vital in the breeding of early-maturity and high-yield crop varieties. An important index to evaluate the early maturity of crops is the time-of-flowering traits, and the expression of genes that determine flowering affects the time of flowering and directly controls the length of the crop growth period [4,5]. Therefore, it is essential to understand the network regulating foxtail millet flowering in order to develop early-maturing varieties and ultimately incorporate them into the planting structure of alpine regions.

The regulation of flowering time in plants is a self-adaptation to the environment and a major determinant of cereal crop yield. The photoperiod regulation of plant flowering has been extensively reported in Arabidopsis and rice [6,7,8,9]. To date, QTL mapping has been used to clone many genes involved in the photoperiod control of flowering pathways in rice, including Hd3a, Hd1, Ehd1, and Ghd7 [10,11,12]. Foxtail millet is widely regarded as a model C4 and energy crop [13,14], and the cloning and functional analysis of millet stress-related genes are increasing [15]. However, there are few studies on the photoperiod, mainly on the QTL mapping of photoperiod-sensitive correlations [16], and few reports on the screening, cloning, and functional identification of photoperiod-related genes.

The GI–CO–FT pathway is involved in photoperiodic responses in both dicots and monocots. The CO gene is a key gene in the photoperiodic response because it integrates light and clock signals, regulates interactions between various proteins, and thus plays a role in inhibiting or promoting flowering [10,17,18]. FLOWERING LOCUS T (FT), which belongs to the phosphatidylethanolamine binding protein (PEBP) family, controls flowering time by encoding a small protein called florigen. The CO-induced initiation of FT and other downstream genes must occur in combination with nuclear factor-Y (NF-Y) [19,20,21]. In an NF-Y transcription factor mutant, FT expression decreases [22]. Nuclear factor-Y is a class of transcription factors that are widely distributed in eukaryotes and are also known as heme-activator proteins (HAPs) or CCAAT-binding factors (CBFs). Nuclear factor-Y usually regulates the expression of downstream genes in the form of heterotrimers composed of three subunits: NF-YA (CBF-B/HAP2), NF-YB (CBF-A/HAP3), and NF-YC (CBF-C/HAP5) [23]. The transcription and protein levels of CO and NF-Y transcription factors change with time, and thus, the regulation of FT by NF-YA/B/C and NF-YB/C-CO trimers changes regularly to dynamically regulate the plant flowering process. The overexpression of a single NF-Y gene can alter plant flowering time [24], suggesting that NF-Y subunits are involved in regulating flowering time in a highly redundant and complex mechanism. However, how the CO gene regulates FT is not well understood.

In a previous study [25], 116 candidate genes were selected by mapping photoperiod-sensitive traits in millet, among which 4 were members of the NF-Y gene family. Studies on the functions of NF-Y family genes are limited to stress resistance [26,27], and little attention has been paid to the regulation of heading and flowering times in millet.

In this study, the candidate gene *SiNF-YC2*, associated with the regulation of flowering in foxtail millet, was cloned, and the biological characteristics and functions were examined. The *SiNF-YC2* gene promoted early flowering and improved plant salt tolerance. The results of this study provide a foundation to study the mechanism regulating the early maturation of foxtail millet and also guidance to improve the molecular breeding of foxtail millet.

## 2. Results

### 2.1. Localization and Bioinformatics Analysis of SiNF-YC2

In our previous study [21], 116 candidate genes were predicted according to gene annotations. The *SiNF-YC2* gene, which is located in the region of Block 63,401–63,475 on chromosome 9 (Figure 1), was identified as an important candidate gene based on the integrated results of gene annotation, cis-element analysis, and the expression patterns of candidate genes in different varieties that have different photoperiod sensitivities.

According to bioinformatics analysis, *SiNF-YC2* is an unstable protein with an open coding frame of 738 bp and 245 amino acid residues. The content of hydrophilic amino acids in the *SiNF-YC2* peptide chain is higher than that of hydrophobic amino acids, indicating that *SiNF-YC2* is a hydrophilic protein (Appendix A). The whole *SiNF-YC2* peptide chain does not have a transmembrane domain, indicating that it is not a transmembrane protein (Appendix A). According to signal peptide prediction, the *SiNF-YC2* protein does not have a signal peptide, is a nonsecreted protein, and likely has multiple phosphorylation sites (Appendix A).

In the analysis of conserved *SiNF-YC2* domains, a conserved HAP5 domain was predicted in the 49–148 amino acid region of the protein, indicating its membership in the CCAAT-box-bound NF-YC transcription factor family (Figure 2a). According to the analysis of the secondary structure, the *SiNF-YC2* protein is composed of 28.98% α-helices (71 amino acids), 11.43% elongation links (28 amino acids), 5.71% β-angles (14 amino acids), and 53.88% random coils (132 amino acids) (Figure 2b). SWISS-MODEL (https://swissmodel.expasy.org/interactive, accessed on 6 October 2022) was used to predict the tertiary structure of the *SiNF-YC2* protein. The similarity between the template sequence required for modeling and the target sequence reached 58.18%, indicating that the prediction results were close to the actual results (Figure 2e).

According to the position of *SiNF-YC2* in the millet genome, the promoter of *SiNF-YC2* was analyzed using the online analysis software PlantCARE(plant cis-acting regulatory elements, http://bioinformatics.psd.ugent.be/welotools/plantcare/html/, accessed on 14 October 2022). In addition to the basic regulatory elements TATA box and CAAT box, the promoter of the millet *SiNF-YC2* gene also contains several hormone-related action elements, such as ABRE, the CGTCA motif, and the TGACG motif; optical response elements, such as the G-box, G-Box, GA motif, MRE, and TCT motif; and anti-stress-response-related elements, such as TC-rich repeats. In addition, the promoter contains ARE action elements that respond to anaerobic induction and RY cis-acting elements that participate in seed-specific regulation (Figure 2c; Appendix A).

The sequence alignment of the *SiNF-YC2* protein with the conserved domain of NF-YC proteins in other species showed that it also contains two β-linked domains separated by α 1, α 2, and α 3 helices and an α C structure, which are indicators of the NF-YC family (Appendix A). In the phylogenetic analysis of NF-YC family members in A. *thaliana*, maize, and rice, *SiNF-YC2* of millet is closely related to *ZmNF-YC1* of maize and *OsNF-YC2* of rice (Figure 2d).

### 2.2. Analysis of Expression Characteristics of SiNF-YC2

To analyze circadian expression, fresh leaf samples from plants growing under SD and LD photoperiods were collected every 3 h starting at 10 a.m. (light onset) for 48 h. The *SiNF-YC2* gene was rhythmically expressed in the leaves of Longgu 3 for 24 h under both long- and short-sunshine conditions (Figure 3a,b). Under the SD photoperiod, the expression level decreased with the onset of light, reached the lowest level at 3 h, then began to increase, and reached the first peak at 6 h. The expression level of *SiNF-YC2* reached a second peak at the 10th hour of darkness, then decreased, and reached the lowest level at the 13th hour of darkness. At the end of darkness, the expression level of *SiNF-YC2* reached a third peak. In the next cycle, 24 h later, the same trend was observed. Thus, the expression of *SiNF-YC2* showed three peaks within one day, one in the light period and two in the dark period (Figure 3a). Under the LD photoperiod, the expression of *SiNF-YC2* also began to decrease with the onset of light and began to increase at 3 h but reached a first peak at 9 h of light, with a second peak at 2 h of darkness (18 h). As was observed under the SD photoperiod, there was a third peak of *SiNF-YC2* expression at the end of darkness. The expression pattern in the next cycle was similar to that in the previous 24 h (Figure 3b). The results showed that the expression pattern of the *SiNF-YC2* gene was different under different photoperiods.

The *SiNF-YC2* gene was expressed in the different tissues of Longgu 3, but the relative expression was significantly different (*p* < 0.05). The relative expression of the *SiNF-YC2* gene was the highest in the flag leaf, followed by that in roots, with the lowest expression in the stem (Figure 3c).

The expression *SiNF-YC2* was examined under different stress treatments (Figure 3d–f). In the 20% PEG6000 treatment, the expression of the *SiNF-YC2* gene was significantly different from that in the control at different times, and the highest expression was at 48 h, which was approximately 2.5 times higher than that in the control. In the NaCl treatment, the expression of the *SiNF-YC2* gene was significantly different from that in the control group, first decreasing and then increasing, then decreasing and then increasing, and finally decreasing. A first peak appeared at 3 h after treatment, and a second peak appeared at 24 h after treatment. The expression of *SiNF-YC2* in the ABA treatment was significantly lower than that in the control, except at 3 h.

### 2.3. SiNF-YC2 Interacts with SiCO in the Nucleus

The localization results of *SiNF-YC2* showed that the p35S-*SiNF-YC2*-GFP fusion protein only emitted a green fluorescence signal in the nucleus, indicating that the *SiNF-YC2* protein was primarily localized in the nucleus (Figure 4a). In the Y2H assays, the experimental group was able to grow on the four-deficient SD/-Ade-His-Leu-Trp medium containing X-α-gal and was stained blue, which indicated that *SiNF-YC2* interacted with *SiCO* (Figure 4b).

### 2.4. Functional Analysis of SiNF-YC2

Leaf DNA was extracted from plants screened on hygromycin medium, and then PCR was performed. Nine of ten transgenic Arabidopsis (OE-1 to OE-10) plants contained the target genes (Figure 5a). Then, three lines with high expression, which were OE-1, OE-3, and OE-5, were screened from the other six lines by RT-qPCR and used for subsequent functional analysis (Figure 5b).

The *SiNF-YC2* gene showed circadian expression patterns in both LD and SD photoperiods (Figure 3a,b), indicating that *SiNF-YC2* was sensitive to the photoperiod and regulated by biological rhythms. Therefore, *SiNF-YC2* was overexpressed in wild-type Arabidopsis plants to study its function. Compared with wild-type Arabidopsis, transgenic Arabidopsis lines overexpressing *SiNF-YC2* produced fewer rosette leaves (Figure 5e) and flowered earlier (Figure 5c,d), indicating that *SiNF-YC2* positively regulated flowering time in Arabidopsis.

Transgenic Arabidopsis performed differently under different stress treatments (Figure 6). At different concentrations of NaCl, the germination rate of transgenic Arabidopsis was slightly higher than that of the wild type, but the difference was not significant (Figure 6g). After seven days of vertical culture, the root length of transgenic *SiNF-YC2* Arabidopsis was not significantly different from that of the wild type on normal medium, whereas the root length of *SiNF-YC2* transgenic Arabidopsis was significantly higher than that of the wild type under 125 mM and 150 mM NaCl stress (Figure 6h,i). After transplanting to soil, Arabidopsis was treated with salt. The results further indicated that *SiNF-YC2* played a role in promoting the response of transgenic Arabidopsis to salt stress (Figure 6c–j,k). At different concentrations of mannitol and ABA, the phenotypic results of transgenic Arabidopsis indicated that there was no difference in sensitivity between *SiNF-YC2* transgenic Arabidopsis and the wild type at either the germination or seedling stage (Figure 6a–f). Thus, the results indicated that *SiNF-YC2* did not have an important role in promoting the response to osmotic stress in Arabidopsis.

## 3. Discussion

A relatively long crop growth period affects the cultivation of the resulting crops, and the yield and quality of late-maturing varieties can also be adversely affected by climatic factors such as high temperatures or heavy rains. In cold areas with cultivation, low-temperature injury is a major threat to crops, primarily affecting the heading, grouting, and seed-setting rate of late-ripening varieties. Although late-ripening varieties can produce high and stable yields, it is difficult to optimize the agricultural industry structure with such varieties. However, early-maturing varieties can effectively avoid the effects of cold, dew, and wind and often achieve high and stable yields [28]. Therefore, early-maturing varieties with an appropriate maturity stage need to be selected. The discovery of early-maturity genes was an important step in selecting crop varieties with early maturity and high yields [28]. *OsNF-YB11* is very important for the heading date, plant height, and yield of rice [29]. *OsNF-YB1* is specifically expressed in the aleurone layer of developing endosperm and regulates grain filling and endosperm development, and it affects grain yield and quality [30,31]. Flowering time is an important index to determine the maturity stage of crops [32]. However, the molecular mechanism regulating flowering in millet remains unclear. The NF-YC protein is an important component of the NF-Y transcription factor complex. In previous studies [25], *SiNF-YC2* was isolated, but its function remained unclear. In this study, the *SiNF-YC2* protein was cloned and analyzed, and it was found to contain DNA-binding domains, NF-YA and NF-YB interaction domains, and CCAAT-binding domains. This study confirmed that the *SiNF-YC2* gene is associated with early maturity, which provides genetic resources and theoretical support for the cultivation of early-maturity millet varieties.

In the present study, our results showed that *SiNF-YC2* may promote early flowering, which is similar to the function of *AtNF-YC4* in *Arabidopsis thaliana*. This means that the function of homologous genes may be conserved; however, this conclusion is not always valid: for example, Hd1, which is the homologous gene of the Arabidopsis CO gene in rice, negatively regulates the FT co-orthologs HD3A and RFT1 under long-day conditions, which means that the function of homologous genes in different plants may vary. In addition, the function of different members of the NF-YC gene family may vary. In *Arabidopsis thaliana*, overexpression of the *AtNF-YC2/3* gene can promote the flowering of plants and increase the transcription level of FT [33]. In rice, *OsNF-YC2* and *OsNF-YC6* can interact with DHD1 (delayed heading date1), a member of the GRAS family, to synergically inhibit the expression of Ehd1, thus delaying the heading date [34]. The above research showed that the overexpression of the NF-Y gene can alter flowering time via a highly redundant and complex mechanism in various plants [35]; therefore, there is still much work to be completed on the function of NF-Y genes in flowering regulation in various plants.

In the present study, our result showed that *SiNF-YC2* interacted with *SiCO* in the nucleus. Previous research showed that both NF-YA/B/C trimers and NF-YB/C-CO (CONSTANS) trimers can be targeted to the promoter region of FT [35,36]. An NF-Y protein can recognize the CCAAT site in the promoter. The CCAAT-acting element in the distal promoter is close to the CORE site, and through the interaction between the NF-Y complex and CO protein, the CO-binding CORE is recruited. The recruitment makes the FT promoter form a ring, which then positively regulates the transcription of FT to promote plant flowering. So, we speculate that *SiNF-YC2* may participate in the flowering regulation of foxtail millet by forming a dimer with the CO protein and then inducing the expression of florin by the FT gene [22]. However, the detailed molecular mechanism of how *SiNF-YC2* binds to FT promoters is not yet clear, so further verification is required in foxtail millet.

According to previous studies, NF-YC functions in the abiotic stress response [27,37,38]. *AtNF-YC1* regulates the response of Arabidopsis to cold stress by binding to the CCAAT box in the promoter region of *AtXTH21* [39]. In rice, *OsNF-YC1* is involved in the response to salt stress, and the salt tolerance of overexpressed *OsNF-YC1* plants increases significantly compared with that of the wild type [40]. In addition, *TaNF-YC5* is induced by drought and salt stress and is important in the resistance to drought and salt stress [41], and the overexpression of *CdtNF-YC1* in rice increases plant sensitivity to salt stress [42]. The cis-element in the *SiNF-YC2* promoter contains ABRE elements that can increase gene expression under drought and high-salt stress, and RT-qPCR results indicated that it was responsive to drought, salt stress, and ABA. In this study, Arabidopsis plants were treated with different concentrations of NaCl, and the salt tolerance of transgenic *SiNF-YC2* plants was higher than that of the control, indicating that *SiNF-YC2* could increase plant salt tolerance to a certain extent. However, the overexpression of *SiNF-YC2* did not increase the resistance of transgenic plants to drought and ABA. The absence of a response to drought and ABA may be because *SiNF-YC2* needs to bind to NF-YB and NF-YA to form a polymer in order to perform its function. Therefore, further studies are needed to determine whether *SiNF-YC2* is involved in regulating plant responses to drought and ABA stresses and identify the possible regulatory mechanisms.

## 4. Materials and Methods

### 4.1. Bioinformatics Analysis of SiNF-YC2

Several candidate genes were screened by QTL mapping in the early stage [21]. Among the candidate genes, *SiNF-YC2* (Seita. 9G468100) was selected as an important gene for the photoperiod response on the basis of the integrated results of gene annotation, cis-element analysis, and expression analyses. The gene and protein sequences of *SiNF-YC2* were downloaded from the Phytozome database (http://www.phytozome.com/, accessed on 1 October 2022). The physical and chemical properties of the *SiNF-YC2* protein were analyzed in Protparam (https://web.express.org/ProtParam/, accessed on 8 October 2022) [43,44]. ProtScale (https://web.expasy.org/protscale/, accessed on 8 October 2022) was used for hydrophilic/hydrophobic analysis. The online software TMHMM (http://www.cbs.dtu.dk/services/TMHMM/, accessed on 8 October 2022) was used for transmembrane structural domain analysis. The amino acid sequence was input to SignalP (http://www.cbs.dtu.dk/services/SignalP/, accessed on 8 October 2022) to predict whether there was a signal peptide structure. The online CDD (https://www.ncbi.nlm.nih.gov/Structure/cdd/wrpsb.cgi, accessed on 10 October 2022) was used to conduct conserved domain analysis. The software Netphos (http://www.cbs.dtu.dk/services/NetPhos/, accessed on 12 October 2022) was used to analyze protein phosphorylation sites. The secondary structure was predicted using PBIL’s SOMPA online website (https://npsa-prabi.ibcp.fr/cgi-bin/npsa_automat.pl?page=npsa_sopma.html, accessed on 12 October 2022). The tertiary structure was predicted using the SWISS-MODEL of Expasy (https://swissmodel.expasy.org/interactive, accessed on 6 October 2022) with amino acid sequence input. According to the rules of amino acid arrangement, the protein tertiary structure model that had the highest overall quality estimation score was built.

From the Phytozome database, the 1500 bp region upstream of the *SiNF-YC2* gene was identified as the promoter. The plant promoter online analysis software PlantCARE (plant cis-acting regulatory elements, http://bioinformatics.psd.ugent.be/welotools/plantcare/html/, accessed on 14 October 2022) was used to analyze the gene promoter, predict cis-acting elements, and visually analyze them in TBtools.

The protein sequence of *SiNF-YC2* was submitted to the Uniprot database (http://www.uniprot.org, accessed on 8 October 2022) for protein sequence BLAST, and corresponding homologous protein sequences of different species were selected. A phylogenetic tree was constructed using the neighbor-joining method in MEGA 11.0 [45], and the obtained genes were compared by analyzing multiple sequences using DNAMAN8.

### 4.2. Plant Materials and Growth Conditions

The foxtail millet variety Longgu 3 was used for the cloning and expression analysis of the *SiNF-YC2* gene. Colombian zero-type (Col-0) wild *Arabidopsis thaliana* was used as a transgenic receptor and control material.

Longgu 3 foxtail millet was sown in flowerpots (10 cm × 10 cm) with field soil and nutrient soil mixed in a 1:1 ratio and cultured in a light incubator. The temperature was 25 °C, the light intensity was 8000 Lx, and the light time was adjusted according to the test treatment. Wild-type or transgenic Arabidopsis plants (ecotype Columbia (Col-0)) were grown in soil or Murashige and Skoog (MS) medium at 25 °C and 70% humidity with a 16 h light/8 h dark photoperiod at a light intensity of 8000 Lx.

### 4.3. Cloning of SiNF-YC2 and Generation of Transgenic Plants

The gene sequence of *SiNF-YC2* was downloaded from the Phytozome database, and the specific primers C2-F1 and C2-R1 were designed with Primer Premier 5.0 (Premier, Canada) (Appendix A). Total RNA of Longgu 3 was extracted and reverse-transcribed into cDNA. *SiNF-YC2* was amplified using cDNA as a template, and then the PCR product was purified and inserted into a pLB vector. Then, the pLB *SiNF-YC2* plasmid was used as a template, and *SiNF-YC2* was amplified by the primers C2-F2 and C2-R2 (Appendix A), inserted into a pCAMBIA1300 vector (abcam, Catalog No. ab275754), and digested by Kpn I and Sma I. The recombination vector was introduced into *Agrobacterium* strain GV3101 (Biomed, Beijing, China).

*Agrobacterium* tumefaciens strain GV3101 (Biomed, Beijing, China) containing the *SiNF-YC2* recombinant vector was transformed into Arabidopsis by using the floral dip method [46]. Transgenic plants were selected by culturing them on plates supplemented with hygromycin (30 mg/L). Transgenic plants were detected at the DNA and transcription levels by PCR and reverse-transcription quantitative PCR (RT-qPCR), respectively. Homozygous transgenic lines of the T3 generation were screened and propagated for further functional analysis.

### 4.4. Expression Analysis of SiNF-YC2

In each pot, 8 to 10 Longgu 3 were planted, for a total of 28 pots. Under a 12 h light/12 h dark photoperiod, seedlings were grown to the second-leaf stage, and then four seedlings with the same level of growth were retained in each pot. For different photoperiod treatments, when foxtail millet reached the three-leaf stage, 18 pots were placed in an incubator with a short-day photoperiod (8 h light/16 h dark), and 10 pots were placed in an incubator with a long-day photoperiod (16 h light/8 h dark). Two pots of foxtail millet were cultured in the short-day incubator for one week, and then tender leaves were used to extract total RNA and clone *SiNF-YC2*. To characterize the expression of *SiNF-YC2*, six pots of foxtail millet in the short-day incubator were cultured to the heading stage, and the roots, stem, leaf sheath, top leaf, secondary top leaf, and spike were collected, with three replicates of each. Foxtail millet in the different photoperiod treatments was sampled after three weeks of culture. To analyze circadian expression, fresh leaf samples were collected at 3 h intervals over a 48 h period from growing plants. All samples were immediately frozen with liquid nitrogen and stored at –80 °C.

Foxtail millet at the 4-leaf stage under the short-day photoperiod was exposed to salt (100 mM NaCl), drought (20% PEG6000), and ABA (100 µM) stress. Drought and salt treatments were applied by irrigation, the ABA treatment was applied by spraying, and the control group received normal watering. There were 20 pots per treatment. After 0, 1, 3, 6, 12, 24, and 48 h of treatment, fresh leaves were collected, and the expression of *SiNF-YC2* was analyzed under different stress treatments. Three biological replicates were collected at each time point. Samples were immediately frozen with liquid nitrogen and stored at –80 °C.

Total RNA was isolated using a Fast Pure^®^Plant Total RNA Isolation Kit (Vazyme, Nanjing, China) according to the manufacturer’s instructions. First-strand cDNA was reverse-transcribed from total RNA using HiScript II Reverse Transcriptase (Vazyme, Nanjing, China) with oligo (dT) as the primer. PCR was performed in a total volume of 10 µL containing 5 µL of 2× ChamQ SYBR qPCR Master Mix (Vazyme, Nanjing, China), 0.2 µL of each gene-specific primer (Appendix A), 1 µL of cDNA, and 3.6 µL of ddH_2_O on a Roche LightCycler 480 real-time PCR machine. Reactions were conducted using the following program: 95 °C for 2 min and 40 cycles of 95 °C for 5 s, 60 °C for 30 s, and 72 °C for 30 s, and fluorescent signals were detected. The standard procedure for the dissolution curve was 95 °C for 5 s; 60 °C for 1 min; 95 °C continuously; and 50 °C for 30 s. Gene expression was calculated by the 2^−∆∆Ct^ method [47], and data were collated and analyzed. Each experiment was performed with three technical replicates. Student’s *t*-tests were used to determine significant differences.

### 4.5. Subcellular Localization

To determine the subcellular location of the *SiNF-YC2* protein, the fusion expression vector p35S-*SiNF-YC2*-GFP was generated by using homologous recombination. Protoplasts were isolated from rice leaf cells and were transformed with the fusion expression vector p35S-*SiNF-YC2*-GFP+p35S-*OsGhd7*-GFP and the control vector GFP+p35S-*OsGhd7*-GFP. Transformed protoplasts were incubated for 16 to 18 h in the dark. Signals from the green fluorescent protein (GFP) were observed using a confocal laser-scanning microscope (FLV1200, Olympus, Tokyo, Japan), which uses lasers to excite fluorescent dyes in the sample and captures the emitted light to produce high-resolution images.

### 4.6. Yeast Two-Hybrid Assays

To analyze the function and regulatory network of *SiNF-YC2*, yeast two-hybrid assays were conducted. Fusion expression vectors pGADT7-*SiCO* and pGBKT7-*SiNF-YC2* were constructed, and then the fusion expression vector pGADT7-*SiCO*+pGBKT7-*SiNF-YC2* and negative control vectors pGADT7+pGBKT7, pGADT7-*SiCO*+pGBKT7, and pGADT7+pGBKT7-*SiNF-YC2* were transformed separately into yeast (Saccharomyces cerevisiae) strain AH109 by using the polyethylene glycol/lithium acetate method. Transformed yeast was first coated on two-deficient SD/-Leu-Trp medium and then on four-deficient SD/-Ade-His-Leu-Trp medium to observe colony growth. Last, yeast in the test and control groups was applied to media containing X-α-gal. Plates with four-deficient SD/-Ade-His-Leu-Trp medium with gal were cultured upside down at 30 °C for three to four days, and then the change in colony color was observed.

### 4.7. Function Analysis of SiNF-YC2

To analyze the germination rate of transgenic Arabidopsis seeds under stress, 50 plump wild-type and 50 T3 generation transgenic Arabidopsis seeds were selected and, after disinfection, were evenly spot-sown in MS solid medium containing 0, 0.5, 1.0, or 2.0 μM ABA; 0, 75, 100, or 125 mM NaCl; or 0, 100, 200, or 300 mM mannitol. After vernalization, seeds were put into a light-temperature incubator for culture, and germination rates were determined at 1, 1.5, 2, 2.5, 3, and 4 days.

To analyze changes in the root length of transgenic Arabidopsis plants under stress, wild-type and T3-generation transgenic Arabidopsis grown to the second-leaf stage on MS solid medium were transferred to MS solid medium containing 0, 3.0, 6.0, or 9.0 μM ABA; 0, 100, 125, or 150 mM NaCl; or 0, 150, 200, or 300 mM mannitol. Plates were placed vertically, and culturing continued. After differential phenotypes appeared, root lengths were analyzed using a root scanner (WINRHIZO proLA2400). Each experiment was performed with three replicates.

To observe the stress resistance of transgenic Arabidopsis in the seedling stage, wild-type and transgenic Arabidopsis grown on MS solid medium for approximately 10 days were transplanted into soil. After two weeks of normal growth, some of the plants were subjected to drought stress by stopping watering. After differential phenotypes appeared, the water supply was restored, and the survival rate of each line was determined one week after the recovery of growth. The other plants were sprayed with 200 mM NaCl solution. After differential phenotypes appeared, the survival rate of each line was determined.

To observe the flowering phenotype of transgenic Arabidopsis, wild-type and transgenic Arabidopsis grown on MS solid medium for approximately 10 days were transplanted to soil and cultured, with 20 to 25 plants per line. The number of days was recorded from the first day of sowing to the first flower, and the number of rosette leaves was counted.

## 5. Conclusions

*SiNF-YC2* was cloned based on QTL mapping. The expression of the *SiNF-YC2* gene exhibited a circadian rhythm. The gene was highly expressed in millet leaf and root tissues and was involved in responses to abiotic stresses such as drought, salt, and ABA. The overexpression of the *SiNF-YC2* gene increased the resistance of transgenic Arabidopsis to salt stress. The overexpression of *SiNF-YC2* also led to the early flowering of transgenic Arabidopsis, suggesting that the gene is involved in regulating plant flowering by promoting early flowering. The results of this study not only lay a foundation for elucidating the flowering regulation network of foxtail millet but also provide excellent genetic resources for the breeding of early-maturing foxtail millet varieties.

## Figures and Tables

**Figure 1 ijms-24-07217-f001:**
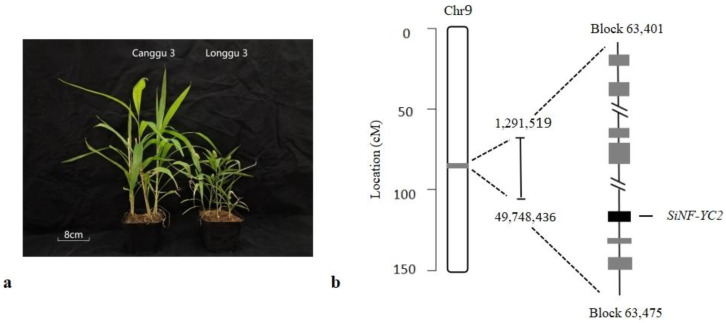
QTL mapping and candidate gene screening for early-maturity traits in foxtail millet. (**a**) Heading phenotype of parent plants. (**b**) Location of *SiNF-YC2* on the chromosome.

**Figure 2 ijms-24-07217-f002:**
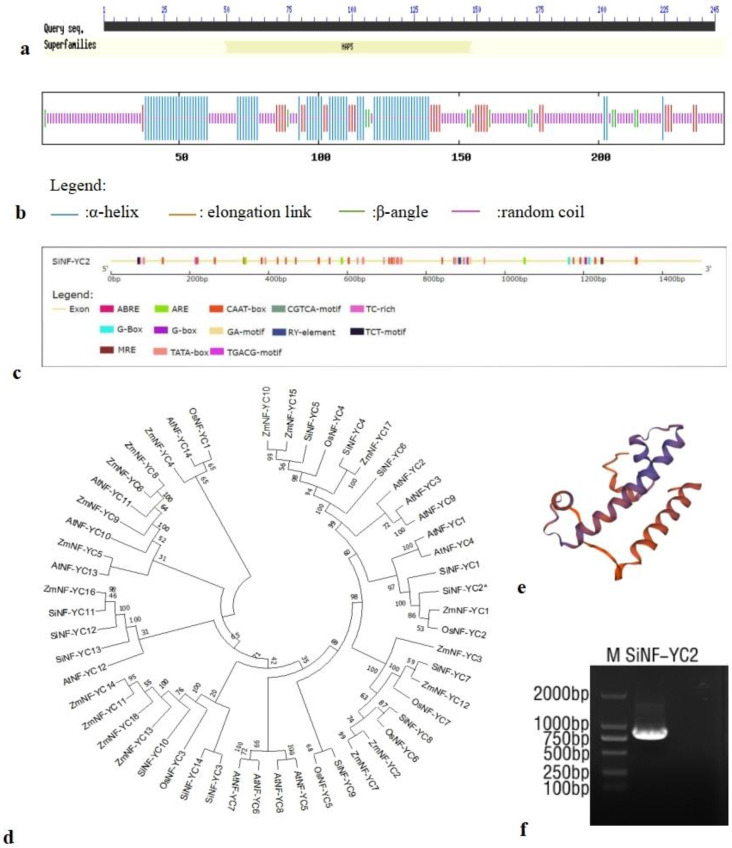
Bioinformatics analysis of *SiNF-YC2*. (**a**) Conserved domain prediction. (**b**) Protein secondary structure analysis. (**c**) cis-Acting elements in *SiNF-YC2* promoter. (**d**) Evolutionary analysis of *SiNF-YC2* (At: *Arabidopsis thaliana*; Zm: *Zea mays*; Si: *Setaria italica*; Os: *Oryza sativa*). The asterisk indicates *SiNF-YC2*. (**e**) Protein tertiary structure prediction. (**f**) Cloning of *SiNF-YC2* (M: 2 kb DNA marker).

**Figure 3 ijms-24-07217-f003:**
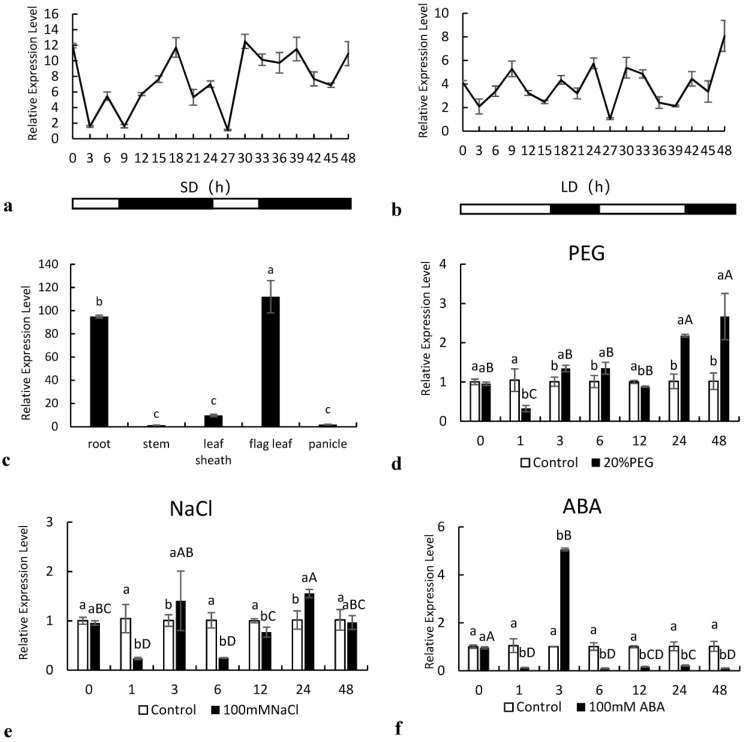
Expression analysis of *SiNF-YC2*. (**a**) Short-day (SD) photoperiod. (**b**) Long-day (LD) photoperiod. (**c**) Relative expression of *SiNF-YC2* in different tissues of foxtail millet. Expression of *SiNF-YC2* under (**d**) 20% PEG6000, (**e**) 200 mM NaCl, and (**f**) 100 μM ABA treatments. Different letters on columns indicate significant differences at the level of 0.05. *p* < 0.05 indicates significant differences between statistical data, *p* > 0.05 indicates insignificant differences between statistical data, and *p* < 0.01 indicates extremely significant differences between statistical data. Different lowercase letters indicate differences between the control and treatment at the same time point; different capital letters indicated differences in SiNF-YC2 expression in millet exposed to stress at different time points.

**Figure 4 ijms-24-07217-f004:**
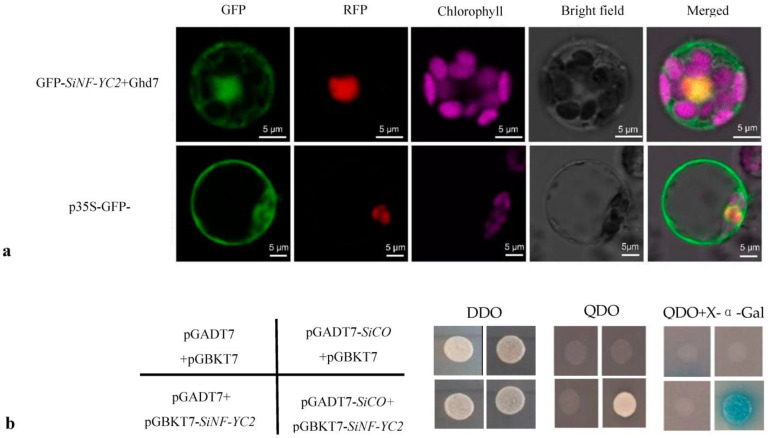
*SiNF-YC2* and *SiCO* interact in the nucleus. (**a**) Subcellular localization of *SiNF-YC2* protein. (**b**) Interaction of *SiNF-YC2* and *SiCO*. DDO: two-deficient SD/-Leu-Trp medium; QDO: four-deficient SD/-Ade-His-Leu-Trp medium.

**Figure 5 ijms-24-07217-f005:**
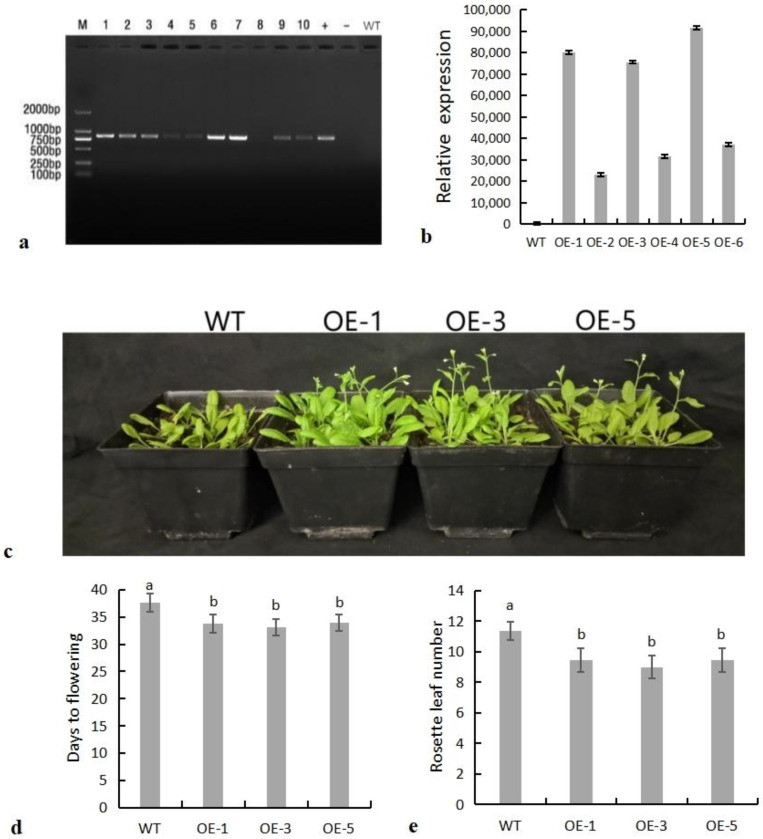
*SiNF-YC2* positively regulates flowering time. (**a**) PCR detection of transgenic Arabidopsis (M: 2 kb DNA marker; +: positive control; −: negative control; WT: wild-type Arabidopsis; 1–10: transgenic Arabidopsis OE-1 to OE-10). (**b**) Quantitative screening of *SiNF-YC2*. (**c**) Flowering phenotypes of wild-type and transgenic Arabidopsis. (**d**) Flowering time statistics. (**e**) Rosette leaf phenotypes of wild-type and transgenic Arabidopsis. Different letters on columns indicate significant differences at the level of 0.05. *p* < 0.05 indicates significant differences between statistical data, *p* > 0.05 indicates insignificant differences between statistical data, and *p* < 0.01 indicates extremely significant differences between statistical data. Different lowercase letters indicate differences between different strains.

**Figure 6 ijms-24-07217-f006:**
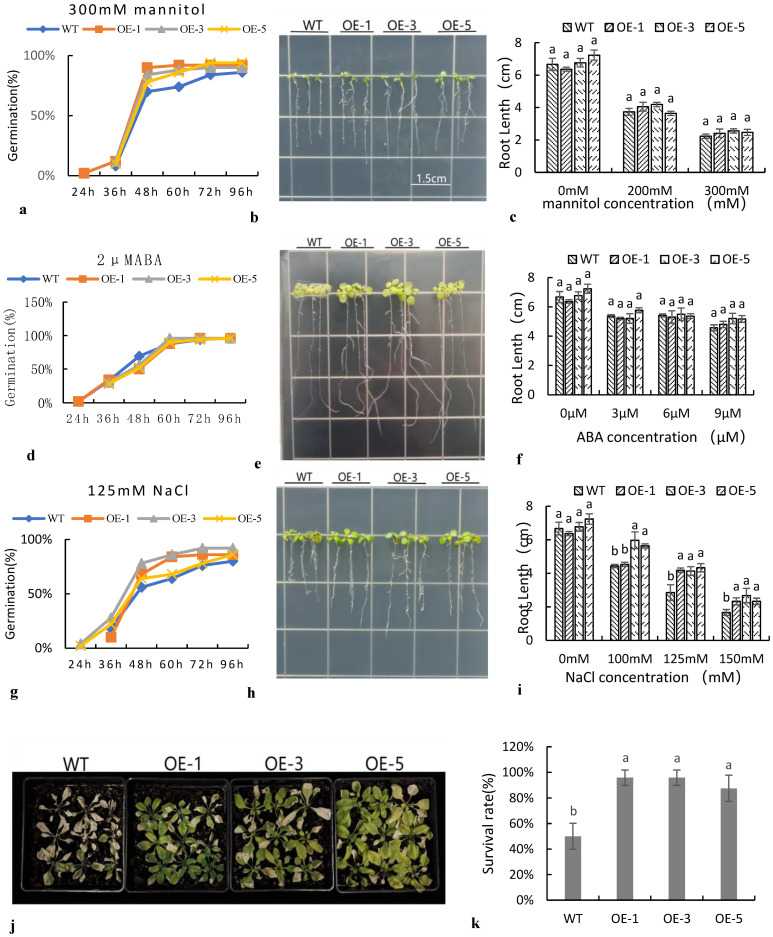
Comparison of transgenic Arabidopsis germination under different stress treatments. Response of transgenic Arabidopsis treated with (**a**–**c**) mannitol, (**d**–**f**) ABA, and (**g**–**k**) salt. (**a**,**d**,**g**) Germination rate, (**b**,**e**,**h**) root length, (**c**,**f**,**i**) root length data analysis, (**j**) salt stress at seedling stage, and (**k**) survival rate. WT: wild type. OE: *SiNF-YC2* transgenic Arabidopsis. Bar = 1.5 cm. Different letters on columns indicate significant differences at the level of 0.05. *p* < 0.05 indicates significant differences between statistical data, *p* > 0.05 indicates insignificant differences between statistical data, and *p* < 0.01 indicates extremely significant differences between statistical data. Different lowercase letters indicate differences between different strains.

## Data Availability

Not applicable.

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
