# Peer review of "SiNF-YC2 Regulates Early Maturity and Salt Tolerance in Setaria italica"

_ijms, 2023, doi:10.3390/ijms24087217_

Round 1
Reviewer 1 Report
Dear Authors,
I am unable to review this article. There is no correct format of the supplementary figure/files. For example: ¦Ô¦¤ŻŠF¬¨Ď˛ÂĘ+¬-ŞĐí¦řżŢ this is one supplementary excel file name, what I downloaded. Also, the content has a lot of chinese characters, so I cannot read it. Please look at the Figure 1 photo too, with chinese characters . There is no Supplementary figure legend. The authors have to correct and re-organise the files, remove all of the chinese characters.
Author Response
Dear Reviewer:
Thank you for your comment concerning our manuscript entitled “SiNF-YC2 regulates early maturity and salt tolerance in Setaria italica”.
1.
I am unable to review this article. There is no correct format of the supplementary figure/files. For example: ¦Ô¦¤ŻŠF¬¨Ď˛ÂĘ+¬-ŞĐí¦řżŢ this is one supplementary excel file name, what I downloaded. Also, the content has a lot of chinese characters, so I cannot read it. Please look at the Figure 1 photo too, with chinese characters . There is no Supplementary figure legend. The authors have to correct and re-organise the files, remove all of the chinese characters.
Thank you for your kindly prompt, we have adjusted the format of the supplementary figure and files, we have modified the chinese to English, and added Supplementary figure legends. You can see the changes in the attachment.

Reviewer 2 Report
The authors studied SiNF-YC2 regulates early maturity and salt tolerance in Setaria italica. They isolated a putative candidate gene, SiNF-YC2 based on quantitative trait loci (QTL) mapping. They found that SiNF-YC2 had a conserved HAP5 domain, which is member of the NF-YC transcription factor family.
My comments are as follows
Line 173: SiNF-YC2 interact with SiCO interacts.
Line 193:(b) Quantitative screening of SiNF-YC2.
In line 173 and 193 it seems that you have quantitative data, if SiNF-YC2 interact with SiCO, what is the direction of the interaction, and can correlation be quantified?
From line 255 to Line 284, you have discussed some agronomic traits in relation to SiNF-YC2 and other candidate genes. In classical breeding early maturity has strong effect on seed yield, plant biomass, see weight etc. You have never discussed these yield related traits in association to the candidate genes.
Author Response
Dear Reviewer:
Thank you for your comments concerning our manuscript entitled “SiNF-YC2 regulates early maturity and salt tolerance in Setaria italica”. Those comments are all valuable and very helpful for revising and improving our paper, as well as the important guiding significance to our researches. We have studied comments carefully and have made corrections which we hope meet with approval. The main corrections in the paper and the responds to your comments are as following:
1.
Line 173: SiNF-YC2 interact with SiCO interacts.
Line 193:(b) Quantitative screening of SiNF-YC2.
In line 173 and 193 it seems that you have quantitative data, if SiNF-YC2 interact with SiCO, what is the direction of the interaction, and can correlation be quantified?
Thank you for your question, according to our results (the yeast two-hybrid assays and phenotypes of transgenic Arabidopsis thaliana) in the present study and the previous reports in Arabidopsis (Kumimoto et al.2010;Xu et al.2016), we think the direction of the interaction is positive, but SiNF-YC2 does not directly regulate the expression of SiCO, because in previous reports in Arabidopsis, the result showed that through interaction between the NF-Y complex and CO protein, the CO-binding CORE is recruited. The recruitment makes the promoter of the FT turn to a ring, which then positively regulates the transcription of FT gene. Our result have showed that SiNF-YC2 interact with SiCO interacts, and In figure 5c, compared with wild-type Arabidopsis, transgenic Arabidopsis lines overexpressing SiNF-YC2 produced fewer rosette leaves (Figure 5e) and flowered earlier, indicated that overexpression of SiNF-YC2 could positively regulate FT transcription and promote plant flowering. So we speculated the SiNF-YC2 participating flowering regulation of foxtail millet may by form a dimer with CO protein, and then inducing expression of the florin gene FT. it can be quantified. The quantization result can also be obtained by detecting the expression level of FT in transgenic Arabidopsis thaliana overexpressed with SiNF-YC2. In line 193:(b) Quantitative screening of SiNF-YC2, The experiment was to determine the expression level of SiNF-YC2 in transgenic Arabidopsis thaliana, the purpose of this experiment was to screen for lines which have the high expression level of SiNF-YC2.
Kumimoto, R.W.; Zhang, Y.; Siefers, N.; Holt, B.F., 3rd. NF-YC3, NF-YC4 and NF-YC9 are required for CONSTANS-mediated, photoperiod-dependent flowering in Arabidopsis thaliana. Plant J. 2010, 63, 379-391.
Xu, M.Y.; Zhu, J.X.; Zhang, M.; Wang, L. Advances on plant miR169/NF-YA regulation modules. Yi chuan = Hereditas. 2016, 38, 700-706.
2.
From line 255 to Line 284, you have discussed some agronomic traits in relation to SiNF-YC2 and other candidate genes. In classical breeding early maturity has strong effect on seed yield, plant biomass, see weight etc. You have never discussed these yield related traits in association to the candidate genes.
Thank you for your good suggest, we have supplemented the discussion section on yield related traits in association to the NF-Y genes. The discussion section can be seen in line 246 to line 249 of the new manuscript. the function of most NF-Y family genes were involved in flowering and stress tolerance, the reports on the relationship between yield-related traits and NF-Y were rarely, at present, we only know OsNF-YB11 and OsNF-YB1 were reported that were involved in yield of rice. In 2022, the function of NF-Y gene family were summarized in the "Advances on the function of NF-Y transcription factors in regulation of plant growth and development"(Xu et al.2022).
Xu, J.; Niu, B.X.; Chen, C. Advances on the function of NF-Y transcription factors in regulation of plant growth and development. Plant Physiology Journal. 2022, 58, 1191-1200.

Reviewer 3 Report
This research investigates SiNF-YC2, a candidate gene for controlling flowering time that belongs to the NF0Y transcription factor family. Gene expression and function studies suggest that this gene is involved in photoperiodic regulation and early flowering. The results are thoroughly discussed and well-supported. Here are some specific comments and questions regarding the manuscript:
1. The most supportive finding in the study is that overexpression of SiNF-YC2 leads to early flowering in Arabidopsis. However, as the experiment was only performed in Arabidopsis, it would be more conclusive to obtain results from foxtail millet, the target crop species.
2. Please provide a brief introduction to the FT gene when it is first mentioned in the Introduction section.
3. The description of "expression patterns of different genes in different varieties" in line 94 is unclear. Please clarify this statement.
4. The legend for Figure 2b is missing, making it unclear which part of the figure represents the helix and which part represents the link. Please provide a detailed legend for this figure.
5. In Figures 3c-f, 5d-e, and 6c, f, i, and k, please provide the exact p-value to indicate the significant level. It is difficult to understand the significance of the letters without an explanation in the legend.
Author Response
Dear Reviewer:
Thank you for your comments concerning our manuscript entitled “SiNF-YC2 regulates early maturity and salt tolerance in Setaria italica”. Those comments are all valuable and very helpful for revising and improving our paper, as well as the important guiding significance to our researches. We have studied comments carefully and have made corrections which we hope meet with approval. The main corrections in the paper and the responds to your comments are as following:
1.
The most supportive finding in the study is that overexpression of SiNF-YC2 leads to early flowering in Arabidopsis. However, as the experiment was only performed in Arabidopsis, it would be more conclusive to obtain results from foxtail millet, the target crop species.
Thank you for your good suggest, we also agree with your opinion. In fact, we have tried transfer the candidate gene SiNF-YC2 into foxtail millet, but so far, we have not succeeded in obtaining the transgenic millet, so the related phenotypes of transgenic millet plants overexpressing SiNF-YC2 will reported in the future studies.
2.Please provide a brief introduction to the FT gene when it is first mentioned in the Introduction section.
Thank you for your kindly prompt, we have provide a brief introduction on the FT gene. The sentence“ FLOWERING LOCUS T (FT), which belongs to the phosphatidylethanolamine binding protein (PEBP) family, controls flowering time by encoding a small protein called florigen.” can be seen in the new manuscript.
3.The description of "expression patterns of different genes in different varieties" in line 94 is unclear. Please clarify this statement.
The description of "expression patterns of different genes in different varieties" has been modified into “the expression pattern of candidate genes in different variety which have different photoperiod sensitivity”. The sentence can be seen in the new manuscript.
4.The legend for Figure 2b is missing, making it unclear which part of the figure represents the helix and which part represents the link. Please provide a detailed legend for this figure.
According to your suggest,we have provided a detailed legend for the Figure 2b, it can be seen in the new manuscript.
5. In Figures 3c-f, 5d-e, and 6c, f, i, and k, please provide the exact p-value to indicate the significant level. It is difficult to understand the significance of the letters without an explanation in the legend.
Thank you for your kindly prompt, we have provided the exact p-value to indicate the significant level. The sentence “If P<0.05 indicates significant differences between statistical data, P>0.05 indicates insignificant differences between statistical data, and P<0.01 indicates extremely significant differences between statistical data. Different lowercase letters indicate differences between control and treatment at the same time point; Different capital letters indicated the difference of SiNF-YC2 expression in millet treated with stress at different time points” can be seen in the new manuscript.

Round 2
Reviewer 1 Report
In this manuscript, Jiahong Niu (ijms-2288346) and colleagues studies the SiNF-YC2 gene function in Setaria italica and Arabidopsis.
The authors corrected my previous suggestion, but stills I have several questions.
Please use italic throughout the article, for Agrobacterium or Arabidopsis.
Line 129: 2kb DNA marker, the kilo is always small letter
Line 147: Please add the time intervals for LD and SD. It would be good to see here, to understand the Figure 3. properly.
Line 164: SD and LD please add in bracket after words short-day or long-day
On Figure 3: Please add a description, what is it CK? It would be good to use control or mock
Line 180: please remove: - from 100-mM
Line 197: Which plant lines used in this study?
Line 199: Please replace nine strains to nine lines.
Line 204: Small letter of k. Please use 2kb DNA marker. Figure 5. panel a,: please specify which plant lines used in this study.
Line 234: The scale bar is not correct. Please look at the Figure 6. panel b.
Materials methods: Please add the used methods and vectors references.
Line 402: Please add a complete description about the LSM microscopy.
Author Response
Dear Reviewer:
Thank you for your comments concerning our manuscript entitled “SiNF-YC2 regulates early maturity and salt tolerance in Setaria italica”. Those comments are all valuable and very helpful for revising and improving our paper, as well as the important guiding significance to our researches. We have studied comments carefully and have made corrections which we hope meet with approval. The main corrections in the paper and the responds to your comments are as following:
1.
Please use italic throughout the article, for Agrobacterium or Arabidopsis.
Thank you for your kindly prompt, We have corrected italic throughout the article, for Agrobacterium or Arabidopsis. It can be seen in the new manuscript.
2.
Line 129: 2kb DNA marker, the kilo is always small letter
Thank you for your kindly prompt, We have corrected "2K DNA marker" to "2kb DNA marker" in line 129. It can be seen in the new manuscript.
3.
Line 147: Please add the time intervals for LD and SD. It would be good to see here, to understand the Figure 3. properly.
Thank you for your kindly prompt, we have provide a brief description about the time intervals for LD and SD, The sentence “To analyze circadian expression, fresh leaf samples from plants growing under the SD and LD photoperiod were collected every 3 hours starting at 10 a.m. (light onset) for 48 hours.” can be seen in the new manuscript.
4.
Line 164: SD and LD please add in bracket after words short-day or long-day
Thank you for your kindly prompt, We have indicated SD and LD after words short-day or long-day in line 164. It can be seen in the new manuscript.
5.
On Figure 3: Please add a description, what is it CK? It would be good to use control or mock
According to your suggest,we have used “control” to represent control treatment in Figure 3. It can be seen in the new manuscript.
6.
Line 180: please remove: - from 100-mM
Thank you for your kindly prompt, We have removed the 100-mM in line 180.
7.
Line 197: Which plant lines used in this study?
Nine of 10 transgenic Arabidopsis(OE-1 to OE-10) plants contained target genes . Then, three lines with high expression were screened from the former six lines by RT-qPCR, which were OE-1, OE-3, and OE-5 and used for later superficial identification test.
8.
Line 199: Please replace nine strains to nine lines.
Thank you for your kindly prompt, We have corrected strains to lines.
9.
Line 204: Small letter of k. Please use 2kb DNA marker. Figure 5. panel a,: please specify which plant lines used in this study.
Thank you for your kindly prompt, We have corrected "2K DNA marker" to "2kb DNA marker" in line 205 and specifed that OE-1、OE-3 and OE-5 were used in this study. Detailed description can be seen in line 198 to 202 for new manuscript.
10.
Line 234: The scale bar is not correct. Please look at the Figure 6. panel b.
Thank you for your kindly prompt, We have corrected the scale bar in Figure 6. panel b. It can be seen in the new manuscript.
11.
Materials methods: Please add the used methods and vectors references.
According to your suggest,we have provided the used methods and vectors references. The references can be seen in the new manuscript.
12.
Line 402: Please add a complete description about the LSM microscopy.
Thank you for your kindly prompt, we have provided a brief description about the LSM microscopy. The sentence “Signals from the green fluorescent protein (GFP) were observed using a confocal laser-scanning microscope (FLV1200, Olympus, Tokyo, Japan), which uses lasers to excite fluorescent dyes in the sample and captures the emitted light to produce high-resolution images.” can be seen in the new manuscript.
